# Epidemiology of secondary infection after snakebites in center-west Brazil

**João Victor Soares Coriolano Coutinho**[1,2,3]*, **Taiguara Fraga Guimarães**[1,2], **Bruno Borges Valente**[2], **Lísia Gomes Martins de Moura Tomich**[1,2,3,4]

**1** Institute of Tropical Pathology and Public Health, Federal University of Goiás, Goiânia, Goiás, Brazil, **2** State Hospital for Tropical Diseases (HDT), Goiânia, Goiás, Brazil, **3** Hospital das Clínicas (HC) of the Federal University of Goiás (UFG), Goiânia, Goiás, Brazil, **4** University of Rio Verde, Aparecida de Goiânia campus, Goiânia, Goiás, Brazil

\* jvevcmedico@gmail.com

## Abstract

### Background

Snakebites represent a significant health problem in tropical countries, with an annual incidence of 2.7 million cases worldwide. The incidence of secondary infections after snake bites is also high and is usually caused by bacteria from the oral cavity of snakes. *Morganella morganii* has been identified as an important cause of infections and has been guiding antibiotic therapy in several regions of Brazil and the world.

### Methodology/Principal findings

We performed a retrospective cross-sectional evaluation of snakebites in hospitalized patients between January 2018 and November 2019 and selected those with secondary infection in their medical records. During the period, 326 cases of snakebites were treated, and 155 (47.5%) of them eventually had secondary infections. However, only seven patients underwent culture of soft tissue fragments, in which three cases were negative culture results, while *Aeromonas hydrophila* was identified in four cases. Of these, 75% were resistant to ampicillin/sulbactam, 50% had intermediate sensitivity to imipenem, and 25% had intermediate sensitivity to piperacillin/tazobactam. Trimethoprim/sulfamethoxazole (TMP-SMX) was not tested on any strain. Of the 155 cases that progressed to secondary infections, 48.4% (75) were empirically treated with amoxicillin/clavulanate, 41.9% (65) with TMP-SMX, and 32 (22%) of these 144 cases required a change to a second regimen, and 10 of these 32 patients required a third therapeutic regimen.

### Conclusion

Wild animals act as reservoirs of resistant bacteria because their oral cavity favors biofilm formation, which explains the finding of *A. hydrophila* with a reduced sensitivity profile in this study. This fact is essential for the appropriate choice of empirical antibiotic therapy.

**Data Availability Statement:** All relevant data are within the paper, its Supporting Information files

and at https://osf.io/73y84/?view_only=9ae8da2a674544eb9da463c723ebbe35.

**Funding:** The authors received no specific funding for this work.

**Competing interests:** The authors declare no conflict of interest.

## Author summary

Secondary infection is among the most common complications that may occur after an ophidian accident. The associated microorganisms are those found in the oral cavity of snakes, with the bacteria *Morganella Morganii* being described with the highest prevalence. In the oral cavity of snakes, bacteria are protected and can develop resistance to the most common antibiotics used in clinical practice. Our study aimed to study cases of secondary infection after snakebites in a specialized service in Midwest Brazil. The investigation of which bacteria caused the infection was carried out in some cases, and only the bacteria *Aeromonas hydrophila* was isolated, with resistance to the antibiotic most commonly used in the service, which was Amoxicillin with Clavulanate. This led us to discuss the best therapeutic approach for this germ and also for ophidian accidents that complicate with secondary infection in general.

## Introduction

Snakebites are a major public health problem in tropical countries, with approximately 2.7 million accidents annually worldwide [1,2]. In 2020, 29,925 cases of snake accidents were reported in Brazil, with 2,835 (9.5%) of the cases in the midwest region. Of these, 1,261 (44.5%) cases occurred in the State of Goiás [3].

Secondary infection is a frequent complication of snakebites occurring predominantly with *Bothrops* species due to the local inflammatory action of their venom, which favors bacterial proliferation through the inoculation of the oral microflora of the snake [4,5].

Gram-negative and anaerobic bacteria are the most frequently identified causative agents in secondary infections, especially *Morganella morganii*, leading healthcare providers to administer empirical antibiotic therapy based on this evidence [5–7]. However, resistant strains have emerged. Wild animals act as reservoirs of these bacteria due to biofilm formation in their oral cavity and disposal of waste containing heavy metals in river waters, which promotes the selection of resistant bacteria, especially *Aeromonas hydrophila*, a pathogen that adapts quickly and has several virulence mechanisms [8–10].

The study aimed to evaluate the incidence of secondary infections, to characterize the microbiological profile and empirical therapy failure rates.

## Methods

### Ethics statement

The Ethical Committee Board of the institution evaluated and approved the study under protocol number (CAE) 26164619.4.0000.0034. Patient consent was not required, considering the retrospective study design used secondary data.

### Study design

This retrospective cross-sectional study included patients with snakebites hospitalized and treated at the State Hospital for Tropical Diseases (HDT) in Goiânia, Goiás, Brazil, from January 2018 to November 2019. All cases were selected from the notification records of the Hospital Center for Epidemiological Surveillance. Clinical variables of the patients were obtained from medical records.

In our routine, diagnosis and classification of snakebites are frequently performed considering clinical and epidemiological manifestations reported by victims. Identifying the

venomous snake species is not a standard procedure due to the risks of trying to capture, transport, or manipulate these animals. Besides, wildlife conservation laws for animals in Brazil discourage people from killing or capturing them.

According to the severity of the clinical manifestations presented by the victims, the accidents were classified and treated as mild, moderate, or severe, following recommendations of the Brazilian Health Ministry[5]. It is possible that, during de course of treatment of the envenomation, patients may need to receive additional doses of antivenom.

Our criterion to define a secondary infection was the description in the medical records of signs and suggestive symptoms that were associated with a need for antibiotic therapy. Until the current study, performing cultures of tissue fragments for microbiological confirmation of secondary infections was not usually part of our routine at the institution. We observed that in the cases in which this procedure was performed (n = 7), the patients required a surgical approach to drain the abscess, debridement, or fasciotomy—with a specialized general surgery service. The collection of a skin fragment for culture was performed after indication and request from the attending physician when there was no response to the instituted antibiotic therapy.

## Statistical analysis

Data were recorded in a Microsoft Excel spreadsheet and analyzed with STATA 16.1 statistical software (StataCorp, Texas, USA). Distribution was presented as histograms, and categories were described as absolute and relative frequencies. Associations were considered using the Chi-square test or Fisher exact test, and a p-value less than .05 was considered significant.

## Results

A total of 326 cases of snakebite were analyzed in the period (see S1 and S2 Tables), with 47.5% (155) secondary infections observed. Despite the high incidence of this complication, only seven subjects underwent soft tissue fragment culture in which three cases were negative results. In contrast, four cases of *Aeromonas hydrophila* were identified: three *Aeromonas hydrophila* subsp. *caviae*, and one was identified as *A. hydrophila* without subspecies identification (Fig 1).

When we compare the patients who evolved with classified secondary infection with those who did not, we felt there was a significant association for infection in individuals who were more severely affected, victims of bothropic accident (Table 1).

Cephalosporins, aminoglycosides, and quinolones preserved sensitive antimicrobial susceptibility in all strains (Table 2). TMP-SMX was not tested by the laboratory on any of these strains.

Table 3 includes the characteristics of the four individuals with secondary infection and positive culture considered for this analysis.

Regarding the patient with strain 1, the incident occurred at their home. The patient's examination results were abnormal close to the start of antibiotic therapy: blood count of 10,400 leukocytes/mm3 (reference range: 5,000–10,000/mm3), urine type I with 25,000 leukocytes per ml (reference range: < 10,000 leukocytes/ml), and altered clotting time of 13 minutes (reference range: 4–10 minutes). The secondary infection was diagnosed 72 hours after the patient was admitted to our service.

The incident with the patient with strain 2 also occurred at their home. The following examination results were abnormal close to the start of antibiotic therapy: blood count of 16,500 leukocytes/mm3 (reference range: 5,000–10,000/mm3), creatine phosphokinase of 1,124 U/L (reference range: 26 to 155 U/L), lactic dehydrogenase of 675 IU/L (reference range:

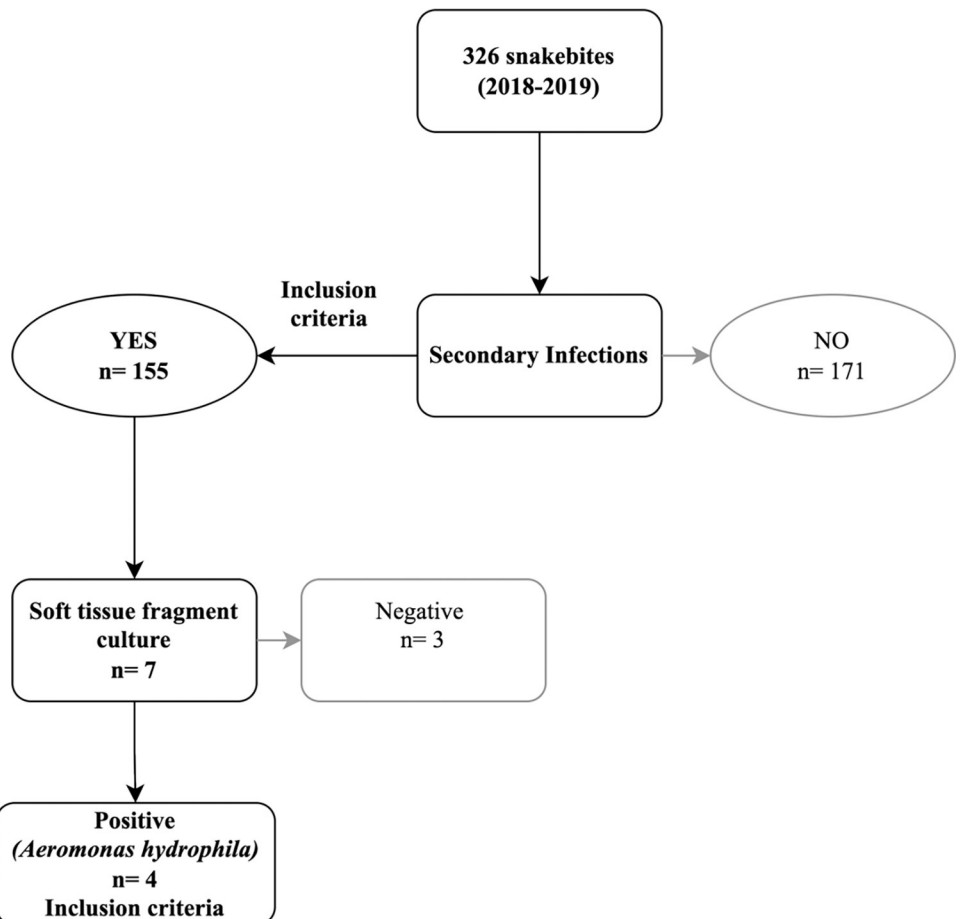

**Fig 1. Flowchart of study inclusion criteria in the analysis of snakebite accidents from 2018 to 2019.**

200 to 480 IU/L) and incoagulable clotting time (reference: 4 to 10 minutes), with a diagnosis of secondary infection established 24 hours after admission to our service.

The patient in strain 3 was from a rural area, and the accident also occurred at their home. The following patient's examination results were altered close to the start of antibiotic therapy: 13,800 leukocytes/mm3 (reference range: 5,000 to 10,000 leukocytes/mm3). The diagnosis of secondary infection was established 24 hours after their admission to our service. The patient's condition developed into osteomyelitis and bone necrosis, verified on computed tomography and radiography of the lower limbs. For these reasons, a transfer to a specialized orthopaedic service was required.

The strain 4 patient was a farmer, and the accident was related to his work. There were no abnormal examination results close to the start date of antibiotic therapy, but local manifestations suggesting secondary infection were described in their medical records, which suggested a need for antibiotic therapy. The diagnosis of secondary infection was established 72 hours after admission to our service.

Of the 155 patients with secondary infection, 48.4% (75) were treated with amox-clav and 41.9% (65) with TMP-SMX. Empirical exchange of antimicrobial regimen occurred in 32 (20.6%) patients: 16 (50%) were initially treated with amox-clav by the physicians; 8 (25%) with TMP-SMX; 3 (9.3%) with ceftriaxone and clindamycin; and 2 (6.2%), cefepime, ciprofloxacin, and clindamycin, or ceftriaxone and oxacillin.

**Table 1. Sample characteristics of patients admitted for snakebites with and without secondary infections.** Hospital for Tropical Diseases, Goiás (Brazil), 2018 to 2019.

| Sample characteristics | Secondary infections | | | | P value |
|---|---|---|---|---|---|
| | Yes (n = 155) | | No (n = 171) | | |
| | n | % | n | % | |
| **Age, mean±SD** | 40,5 ± 20,2 | | 36,9 ± 18,9 | | 0.080 |
| **Gender** | | | | | 0.102 |
| Male | 126 | 81,3 | 126 | 73,7 | |
| Female | 29 | 18,7 | 45 | 26,3 | |
| **Case classification** | | | | | 0.000 |
| Mild | 30 | 19,4 | 77 | 45,0 | |
| Moderate | 71 | 45,8 | 58 | 33,9 | |
| Severe | 54 | 34,8 | 36 | 21,1 | |
| **Admission on ICU** | 10 | 6,5 | 4 | 2,3 | 0.059 |
| **Snake genus** | | | | | 0.000 |
| *Bothrops* | 147 | 94,8 | 121 | 70,7 | |
| Other | 8 | 5,2 | 50 | 29,2 | |

ICU: intensive care unit

*Chi square Test or Exact Fisher Test

The most used second-line regimens were piperacillin/tazobactam in 21.8% (7/32), amox-clav in 18.7% (6/32), and TMP-SMX in 15.6% (5/32) of patients. A third regimen was necessary for 10 (31.2%) patients: piperacillin + tazobactam in three (30%) patients; ciprofloxacin + clindamycin in two (20%); and ceftazidime, cefepime, ceftriaxone + clindamycin, TMP-SMX, or amox-clav in 1 (10%). The third most used drug was piperacillin/tazobactam in

**Table 2. Antimicrobial susceptibility of *Aeromonas hydrophila* specimens (n = 4) identified in wound cultures with secondary infection after a snakebite.** Hospital for Tropical Diseases, Goiás (Brazil), 2022.

| | *Aeromonas hydrophila* | | | |
|---|---|---|---|---|
| | Strain 1[A] | Strain 2[A,C*] | Strain 3[B] | Strain 4[A] |
| **Ampicillin** | - | - | R | R |
| **Ampicillin/sulbactam** | R | - | - | - |
| **Cefoxitin** | S | - | - | - |
| **Ceftazidime** | S | - | S | S |
| **Ceftriaxone** | S | - | S | S |
| **Cefepime** | S | - | S | S |
| **Piperacillin/tazobactam** | I | - | - | - |
| **Imipenem** | I | - | S | I |
| **Meropenem** | - | - | S | S |
| **Amikacin** | S | S | S | S |
| **Gentamicin** | S | S | S | S |
| **Ciprofloxacin** | S | S | S | S |
| **Tigecycline** | S | - | - | - |

Automated sensitivity testing by Vitek-2. R: Resistant; I: Intermediate; S: Sensitive.

[A]CSLI 2018 interpretation

[B]CLSI 2017 interpretation

[C]*A. hydrophila* strain without subspecies identification.

**Table 3. Clinical and epidemiological characteristics of the four patients with bacterial isolation from tissue fragments of patients with complicated secondary infection.** Hospital for Tropical Diseases, Goiás (Brazil), 2018 to 2019.

| Characteristics | Strain 1 | Strain 2 | Strain 3 | Strain 4 |
|---|---|---|---|---|
| Year of occurrence of the snakebite | 2019 | 2018 | 2018 | 2018 |
| Sex | Male | Male | Male | Male |
| Age | 49 years | 42 years | 45 years | 34 years |
| Hometown location | Trindade/GO | Hidrolândia/GO | Bela Vista de Goiás/GO | Goiânia/GO |
| Time elapsed between snakebite and medical care | 12–24 hours | 1–3 hours | 1–3 hours | 1–3 hours |
| Snakebite site | Leg | Leg | Leg | Foot |
| Snake genus | *Bothrops* | *Bothrops* | *Bothrops* | *Bothrops* |
| Case classification | Moderate | Severe | Severe | Moderate |
| Local symptoms | Pain and edema | Pain and edema | Pain, edema, necrosis, and ecchymosis | Pain and edema |
| Local complications | Secondary infection with abscess | Secondary infection with cellulitis and compartment syndrome | Extensive necrosis, secondary infection with abscess and osteomyelitis | Secondary infection with cellulitis and compartment syndrome |
| Systemic complications | None | KDIGO I acute kidney injury | KDIGO I acute kidney injury, sepsis, pulmonary thromboembolism | None |
| Antivenom | BAV, 8 vials | BAV, 12 vials | BAV, 9 vials | BAV, 6 vials |
| Antibiotic regimen 1 | Amox-clav for 10 days | Amox-clav for 15 days | Amox-clav for 3 days | TMP-SMX for 3 days |
| Antibiotic regimen 2 | Not necessary | Not necessary | Piperacillin/tazobactam for 6 days | Cefepime for 7 days |
| Antibiotic regimen 3 | Not necessary | Not necessary | Meropenem for 10 days | Ciprofloxacin + clindamycin for 17 days |
| Surgical approach | Drainage of abscess | Fasciotomy | Debridement and drainage of abscess | Fasciotomy |
| Outcome | Cure | Cure | Transfer | Cure |

Amox-clav: Amoxicillin/clavulanate; KDIGO: Kidney Disease Improving Global Outcomes; BAV: *Bothrops* antivenom; TMP-SMX: trimethoprim/sulfamethoxazole.

4/10 (40%) cases, followed by ciprofloxacin + clindamycin in 2/10 (20%) cases, and TMP-SMX also in 2/10 (20%) cases.

## Discussion

Secondary infection is the most common complication of snakebites. It is mainly related to the *Bothrops* genus due to the proteolytic action of their venom, which provides a favorable environment for bacterial growth at the bite site [4,5,11]. Bacterial isolates from infected wounds are compatible with the oral flora of snakes, with the following bacterial species described in the literature in order of importance: *Morganella morganii*, *Escherichia coli*, *Staphylococcus aureus*, *Providencia* sp., *Enterobacter* sp., *Pseudomonas aeruginosa*, *Proteus* sp., and *Klebsiella* sp. [4,5].

In this study, 47.5% (155/326) of patients developed secondary infections, but only seven underwent skin fragment culture, with *A. hydrophila* being the only bacterium recovered. Although *A. hydrophila* is not commonly related to these infections, this occurrence has been documented in Brazil [4], Asia [12], and Central America [13], with rare studies on its antimicrobial susceptibility profile regarding snakebites.

Wild animals act as reservoirs of resistant bacteria, particularly *A. hydrophila* [9]. The gastrointestinal tract of these animals favors the induction of resistance by facilitating the adhesion of the bacteria and mainly by biofilm formation [9]. In addition, changes in ecosystems, especially in river water components, caused by metals from industries and mining act as selective agents for resistant *A. hydrophila* [8].

Beta-lactam antibiotics will be the most affected by the resistance mechanisms developed by *A. hydrophila* due to high beta-lactamase production, especially class B Metallo-β-lactamase, class C cephalosporinase, and class D penicillinase, which affects the efficacy of penicillin and first-generation cephalosporins. Furthermore, except for *Aeromonas trota* subspecies, all the others are intrinsically resistant to ampicillin [10]. Thus, the antimicrobial choice for this microorganism should include extended-spectrum cephalosporins, monobactams, carbapenems, aminoglycosides, or quinolones [10,14].

Several studies demonstrated the high effectiveness of quinolones and fluoroquinolones for the treatment of *A. hydrophila* infections, but with increasing reports of fish infected with quinolone-resistant *A. hydrophila* and resistant species isolates in water samples [15]. In addition, there are reports of quinolone-resistant *A. hydrophila* in infected wounds of patients undergoing therapy with leeches colonized by this bacterium [16,17]. In this present study, all strains were sensitive to ciprofloxacin.

TMP-SMX is one of the therapeutic options for *A. hydrophila* [18]. However, studies have shown varied antimicrobial sensitivity depending on isolated subspecies, such as *A. hydrophila* subsp. *sobria* and *A. hydrophila* subsp. *caviae*. It also depends on the geographic area because strains from the United States have been more susceptible than isolates from Taiwan [17,18]. The present study identified three isolates of *A. hydrophila* subsp. *caviae*, but TMP-SMX was not tested in our laboratory, which leads us to infer that its empirical use should be carefully evaluated.

TMP-SMX becomes an attractive option for the treatment of secondary infections related to snakebites in developing countries due to its low cost, the possibility of being done orally, and also because it has activity against *Morganella Morganii* (the main pathogen described in secondary infections of snakebites) and Gram-positive bacteria [17,18]. However, to be used empirically for *A. hydrophila*, more regional studies evaluating its sensitivity need to be conducted, and it may be considered in cases of mild infections [17,18].

Antibiotic therapy was changed in 21% (32) of patients who predominantly used amoxclav or TMP-SMX, suggesting a possible occurrence of more cases of *A. hydrophila*. However, the diagnosis of infection in this series considered only the medical records. The *Bothrops* venom components may lead to local inflammation signs that may be mistaken for infection and, therefore, the initial antimicrobial regimen may have been inadequately indicated. A better etiological investigation through lesion fragment culture is necessary for suspected cases of secondary infection to determine the real incidence of *A. hydrophila* in complicated snakebites.

In contrast, a study conducted by Andrade et al. [6] in the same institution as our study between 1984 and 1988 aimed at identifying bacteria associated with secondary infection in snakebites and showed no cases of *A. hydrophila* in 77 positive cultures.

The main limitations of this study include the retrospective design and its small sample with an etiological investigation.

But, in summary, secondary infections, like other types of community-acquired infections related to snakebites, have also changed in recent decades, revealing different bacterial species and more restricted therapeutic possibilities. This study corroborates this assertion by showing the occurrence of *A. hydrophila* with a different sensitivity profile. Thus, the antimicrobial susceptibility surveillance of each region is necessary for a better choice of antibiotic therapy in snakebite cases, in addition to avoiding delays between the bite event and the administration of the antivenom through its wide distribution.

## Supporting information

**S1 Dataset. Data Excel spreadsheet.** The dataset and metadata can be found here: https://osf.io/73y84/?view_only=9ae8da2a674544eb9da463c723ebbe35.
(XLSX)

**S1 Strobe. STROBE.**
(DOCX)

**S1 Table. Sociodemographic characteristics of 326 patients treated for snakebites at the State Hospital for Tropical Diseases, Goiás, Brazil, from 2018 to 2019.**
(DOCX)

**S2 Table. Characterization of clinical presentations on admission and associated complications according to snakebite type.**
(DOCX)

## Author Contributions

**Conceptualization:** João Victor Soares Coriolano Coutinho, Lísia Gomes Martins de Moura Tomich.

**Data curation:** João Victor Soares Coriolano Coutinho, Lísia Gomes Martins de Moura Tomich.

**Formal analysis:** João Victor Soares Coriolano Coutinho.

**Investigation:** João Victor Soares Coriolano Coutinho, Taiguara Fraga Guimarães, Bruno Borges Valente, Lísia Gomes Martins de Moura Tomich.

**Methodology:** João Victor Soares Coriolano Coutinho, Lísia Gomes Martins de Moura Tomich.

**Project administration:** João Victor Soares Coriolano Coutinho, Lísia Gomes Martins de Moura Tomich.

**Supervision:** Lísia Gomes Martins de Moura Tomich.

**Writing – original draft:** João Victor Soares Coriolano Coutinho, Lísia Gomes Martins de Moura Tomich.

**Writing – review & editing:** João Victor Soares Coriolano Coutinho, Lísia Gomes Martins de Moura Tomich.

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
