## [Decision Letter · Decision Letter 0]

14 Oct 2022

Dear Mr Soares Coriolano Coutinho,

Thank you very much for submitting your manuscript "Epidemiologia da infecção secundária após acidentes ofídicos no centro-oeste do Brasil" for consideration at PLOS Neglected Tropical Diseases. As with all papers reviewed by the journal, your manuscript was reviewed by members of the editorial board and by several independent reviewers. In light of the reviews (below this email), we would like to invite the resubmission of a significantly-revised version that takes into account the reviewers' comments. 

The decision of MAJOR REVISION is based on valuable information being in the manuscript but major rewriting, organization, focus and shortening. Note is made that the title of the submitted manuscript was not in English in the website submission but the manuscript title in the document was in English. Please fix that.

We cannot make any decision about publication until we have seen the revised manuscript and your response to the reviewers' comments. Your revised manuscript is also likely to be sent to reviewers for further evaluation.

Sincerely,

Joseph M. Vinetz

Section Editor

Reviewer's Responses to Questions

**Key Review Criteria Required for Acceptance?**

**Methods**

-Are the objectives of the study clearly articulated with a clear testable hypothesis stated?

-Is the study design appropriate to address the stated objectives?

-Is the population clearly described and appropriate for the hypothesis being tested?

-Is the sample size sufficient to ensure adequate power to address the hypothesis being tested?

-Were correct statistical analysis used to support conclusions?

-Are there concerns about ethical or regulatory requirements being met?

Reviewer #1: Yes the objectives of the study clearly articulated with a clear testable hypothesis stated. The study design appropriate to address the stated objectives.

The population clearly described and appropriate for the hypothesis being tested.

-The sample size sufficient to ensure adequate power to address the hypothesis being tested

The is any concern about ethical or regulatory requirements.

Reviewer #2: - I suggest that the authors consult the STROBE instrument as a checklist and complete the necessary information for the study, especially the information on the methods as they need more details such as the identified variables, sample calculation, form of poisoning diagnosis, snake identification, accident classification , established treatment, definition of secondary infection adopted, among others.

**Results**

-Does the analysis presented match the analysis plan?

-Are the results clearly and completely presented?

-Are the figures (Tables, Images) of sufficient quality for clarity?

Reviewer #1: Yes, the analysis presented math the analysis plan. The results are well presented.

Reviewer #2: - I suggest a flow of case evaluation to understand the path of 326 total patients, exclusion criteria to reach 155 patients in the study.

- the first table should be a clinical characterization of the 155 patients in the study to understand the context.

- Results should have risk factor analysis or comparative infection and non-infection groups to improve outcomes.

**Conclusions**

-Are the conclusions supported by the data presented?

-Are the limitations of analysis clearly described?

-Do the authors discuss how these data can be helpful to advance our understanding of the topic under study?

-Is public health relevance addressed?

Reviewer #1: Yes. The authors identified clearly the study limitations . This is retrospective study over one year with a small amount of patients. The conclusions are supported by the data presented.

Reviewer #2: - after the result there must be the discussion topic and then the conclusion.

**Editorial and Data Presentation Modifications?**

Reviewer #1: This review describes significant results and deserves to be published after some minor revisions.

Reviewer #2: - The scope of the Plos Neglected journal requires studies with a methodological complexity and data analysis given their impact factor.

- The presented manuscript needs more robustness for a complete article, however I believe it has enough data to be submitted as a short communication.

**Summary and General Comments**

Reviewer #1: Review article 

Epidemiology of secondary infection after snakebites in west-central Brazil.

The retrospective study on the epidemiology of secondary infection after snakebites in west-central Brazil is well written and well conducted and contributes significantly to assess the incidence of secondary infections and to characterize the microbiological profile and empirical therapy. Of the 155 cases that progressed to secondary infections, 48.4% (75) were treated empirically 32 with amoxicillin clavulanate and 41.9% (65) were treated with SMT +TMP, with 32 of 33 these 144 cases requiring a change to a second regimen, and ten of these 32 requiring a third regimen. 

Your results are not significantly different from other studies published in the literature.

In the literature, wound infection after a snakebite occurs in 9 to 77 percent of patients.

We recently published several articles on secondary infections and found that the primary bacteria responsible for wound infection were A. hydrophila, M. morganii, S. aureus and P. rettgeri. In our series on secondary infections related to Bothrops lanceolatus (Martinique) and Bothros atrox (French Guiana), the isolated bacteria were Aeromonas hydrophila, Morganella morganii, group A Streptococcus, and group B Streptococcus. Patients were treated empirically with third-generation cephalosporin (or amoxicillin-clavulanate), aminoglycoside, and metronidazole combinations. Outcome was favorable in all patients. we suggested at this time that empiric antibiotic therapy including third-generation cephalosporin should be administered to those patients on hospital admission.

Minor issues: 

1. What other germs have you encountered in the oral cavity of snakes?

2. Line 147-148. Why do you think SMT+TMP is one of the treatment options for A. hydrophila?

3. Do you think that TMS+TMP could be an attractive therapeutic strategy to treat envenomed patients in developing countries affected by A. hydrophila?

Reviewer #2: The manuscript “Epidemiology of secondary infection after snakebites in center-west Brazil” describes a very relevant topic about the complications of snakebites and still much discussed and controversial for the establishment of treatment protocols due to the diversity in the choice of antibiotics today.

However, I suggest authors carefully observe the structure of articles published in Plos Neglected to improve the next submission.

PLOS authors have the option to publish the peer review history of their article (what does this mean?). If published, this will include your full peer review and any attached files.

Reviewer #1: Yes: Dabor Resiere

Reviewer #2: No
---

## [Decision Letter · Decision Letter 1]

2 Jan 2023

Dear Mr Soares Coriolano Coutinho,

Thank you very much for submitting your manuscript "Epidemiology of secondary infection after snakebites in center-west Brazil" for consideration at PLOS Neglected Tropical Diseases. As with all papers reviewed by the journal, your manuscript was reviewed by members of the editorial board and by several independent reviewers. The reviewers appreciated the attention to an important topic. Based on the reviews, we are likely to accept this manuscript for publication, providing that you modify the manuscript according to the review recommendations. 

Sincerely,

Joseph M. Vinetz

Section Editor

Joseph Vinetz

Section Editor

Reviewer's Responses to Questions

**Key Review Criteria Required for Acceptance?**

**Methods**

-Are the objectives of the study clearly articulated with a clear testable hypothesis stated?

-Is the study design appropriate to address the stated objectives?

-Is the population clearly described and appropriate for the hypothesis being tested?

-Is the sample size sufficient to ensure adequate power to address the hypothesis being tested?

-Were correct statistical analysis used to support conclusions?

-Are there concerns about ethical or regulatory requirements being met?

Reviewer #2: In the methodology, it is necessary to have the definition of secondary infection used to establish the diagnosis and treatment in this study.

For the biopsies performed on the 4 patients described, the authors said that this biopsy procedure is not routine so why was it performed on these patients? Was there any indication needed for this?

**Results**

-Does the analysis presented match the analysis plan?

-Are the results clearly and completely presented?

-Are the figures (Tables, Images) of sufficient quality for clarity?

Reviewer #2: Line 102 – the information from table 2 must be after the description from table 1.

The term “Serum therapy” is not used very often, please change it throughout the text to “antivenom”, and change “ampoules” to “vials” as well.

In table 2, patient number 3 was transferred from the unit, could the authors describe what was his outcome after he was transferred, cure or death? What is the reason for the transfer?

Line 125 – Wouldn't the table named Table 2 be in the order Table 3? Check the correction in the text as well.

Some more interesting information could be described from the cases described, such as activities they were doing at the time of the bite, laboratory tests relevant to the secondary infection, appearance of cellulitis and/or abscess, how many days after the bite the infection was established, among other information.

**Conclusions**

-Are the conclusions supported by the data presented?

-Are the limitations of analysis clearly described?

-Do the authors discuss how these data can be helpful to advance our understanding of the topic under study?

-Is public health relevance addressed?

Reviewer #2: (No Response)

**Editorial and Data Presentation Modifications?**

Reviewer #2: (No Response)

**Summary and General Comments**

Reviewer #2: The authors have made good improvements to the information in the previously submitted article. However, some information still needs clarification.

The same information in the abstract needs to be contained in the text, however, I did not find information about the 7 patients who underwent biopsy and that 3 of these had a negative culture, please add it to the methodology.

PLOS authors have the option to publish the peer review history of their article (what does this mean?). If published, this will include your full peer review and any attached files.

Reviewer #2: No

Figure Files:

Data Requirements:

Reproducibility:

References

---

## [Editor Report · Decision Letter 2]

30 Jan 2023

Dear Mr Soares Coriolano Coutinho,

Thank you very much for submitting your manuscript "Epidemiology of secondary infection after snakebites in center-west Brazil" for consideration at PLOS Neglected Tropical Diseases. As with all papers reviewed by the journal, your manuscript was reviewed by members of the editorial board and by several independent reviewers. In light of the reviews (below this email), we would like to invite the resubmission of a significantly-revised version that takes into account the reviewers' comments. 

We cannot make any decision about publication until we have seen the revised manuscript and your response to the reviewers' comments. Your revised manuscript is also likely to be sent to reviewers for further evaluation.

Sincerely,

Joseph M. Vinetz

Section Editor

Joseph Vinetz

Section Editor
---

## [Editor Report · Decision Letter 3]

13 Feb 2023

Dear Mr Soares Coriolano Coutinho,

We are pleased to inform you that your manuscript 'Epidemiology of secondary infection after snakebites in center-west Brazil' has been provisionally accepted for publication in PLOS Neglected Tropical Diseases.

Best regards,

Joseph M. Vinetz

Section Editor

Joseph Vinetz

Section Editor

---

## [Editor Report · Acceptance letter]

1 Mar 2023

Dear Mr Soares Coriolano Coutinho,

We are delighted to inform you that your manuscript, "Epidemiology of secondary infection after snakebites in center-west Brazil," has been formally accepted for publication in PLOS Neglected Tropical Diseases.

Best regards,

Shaden Kamhawi

co-Editor-in-Chief

Paul Brindley

co-Editor-in-Chief
